# Shared Language Erosion: Rethinking Immigrant Family Communication and Impacts on Youth Development

**DOI:** 10.3390/children8040256

**Published:** 2021-03-25

**Authors:** Ronald B. Cox, Darcey K. deSouza, Juan Bao, Hua Lin, Sumeyra Sahbaz, Kimberly A. Greder, Robert E. Larzelere, Isaac J. Washburn, Maritza Leon-Cartagena, Alma Arredondo-Lopez

**Affiliations:** 1Human Development and Family Science, Oklahoma State University, Stillwater, OK 74078, USA; darcey.desouza@okstate.edu (D.K.d.); hua.lin@okstate.edu (H.L.); sumeyra@okstate.edu (S.S.); robert.larzelere@okstate.edu (R.E.L.); isaac.washburn@okstate.edu (I.J.W.); leoncar@okstatemail.okstate.edu (M.L.-C.); alma.arredondo_lopez@okstate.edu (A.A.-L.); 2Human Development and Family Studies, Iowa State University, Ames, IA 50011, USA; jbao@iastate.edu (J.B.); kgreder@iastate.edu (K.A.G.)

**Keywords:** shared language erosion, acculturation, immigrant families, immigrant paradox, communication, parent-child conflict, adolescent development, parenting, bilingualism

## Abstract

In this paper we make the case for Shared Language Erosion as a potential explanation for the negative outcomes described in the immigrant paradox for second- and third- generation immigrants (e.g., declines in physical, mental, and behavioral health). While not negating the important role of cultural adaptation, we posit that parent-child communication difficulties due to a process we are calling Shared Language Erosion is driving the observed affects previously attributed to changes in cultural values and beliefs. Shared Language Erosion is the process during which adolescents improve their English skills while simultaneously losing or failing to develop their heritage language; at the same time their parents acquire English at a much slower rate. This lack of a common shared language makes it difficult for parents and their adolescent children to effectively communicate with each other, and leads to increased parent-child conflict, reduced parental competence, aggravated preexisting flaws in parent-child attachment, and increased adolescent vulnerability to deviant peer influences.

## 1. Introduction

Communication is central to family life; it enables family members to express and share their needs, joys, aspirations, and concerns, as well as to resolve their problems and find help for their challenges [1]. In its simplest form, communication is the verbal and non-verbal exchange of information through which shared meaning is created. However, at a relational level, communication is also the mechanism through which families are constituted and defined as well as the process through which children are influenced and guided [2]. Communication also functions as a symbol of one’s identity by promoting a sense of belonging and connectedness [3]. However, because communication is intertwined so fully with every aspect of human life, we sometimes miss its pervasiveness, importance, and complexity. Communication impacts every component of family life, making it vital to understanding family functioning and adolescent developmental outcomes.

The language(s) one utilizes to communicate have special consideration for immigrant families in particular. For example, immigrants who choose to maintain a heritage language (HL) communicate their connection to their home culture and people. However, when adaptation into a new culture (a process known as acculturation) changes an individual’s proficiency in one or more languages, it can alter a sense of connection to one’s culture and people, including a connection to one’s family. This, in turn, impacts the meaning-making processes that occur between family members. It affects how they define and participate in relationships with each other [4], and the extent to which these relationships buffer against or expose adolescents to environmental risks such as deviant peer influences. Because talk—a form of communication—is a fundamental means by which family relationships are created and sustained [5], speaking the same language is essential to facilitate intersubjectivity, or the shared thoughts and feelings (both conscious and unconscious) through which families co-construct their reality [6,7].

Studying the impact of family communication on the development of immigrant adolescents may help explain the notorious decline in physical, mental, and behavioral health in second- and third-generation immigrants relative to first-generation immigrants, something that has been termed the immigrant paradox [8]. A growing body of research points to communication difficulties between immigrant parents and their adolescent children (e.g., [9,10,11,12,13]), but the impact of these difficulties on adolescent development remains an understudied area of research. A new way of conceptualizing how family risk and resilience might interact to impact adolescent development in immigrant families is a phenomenon we are calling Shared Language Erosion. In Shared Language Erosion, the developing adolescent becomes increasingly adept and comfortable speaking English due to continuous exposure to school, social and mass media outlets, and by communicating with siblings and friends [14,15]. Simultaneously, growth in the adolescents’ ability to speak their HL is stunted due to a lack of continuous use and to exposure to new domains of knowledge beyond the home [16]. Their parents, on the other hand, tend to increase in their own English language ability at a much slower rate while maintaining their HL [17]. This results in an erosion of a shared language over time, in which parents maintain proficiency in their HL and develop only limited English language skills, and their children develop proficiency in English and inadequately develop and/or lose much of their HL skills.

The purpose of this paper is to make the case that Shared Language Erosion is a potential explanatory factor in previously documented, but not fully explained pressures on immigrant youth development that have resulted in negative outcomes and health disparities, especially after the first generation. Drawing on research from the field of Human Development and Family Science and the fields of Communication and (Applied) Linguistics, we argue that it is the inability of parents and children to effectively communicate with one another, more than previously postulated explanations such as discrepancies in cultural beliefs, values, and behaviors, that help to explain documented declines in the physical, emotional, and behavioral health among immigrant youth as they acculturate to the United States (US). We posit four impacts of Shared Language Erosion on immigrant families: (a) parent-child relationships are damaged due to linguistic and cultural misunderstandings that lead to conflict and frustration, (b) parents’ ability to communicate their life wisdom and to effectively monitor and discipline their developing adolescent is limited, and (c) preexisting deficiencies in parent-child attachment are aggravated. As a result, (d) adolescents in families with Shared Language Erosion are more vulnerable to deviant peer influences and other environmental risks. We also discuss new and exciting areas for research that hold promise to increase understanding of an understudied and underserved population, which can be translated into effective prevention strategies to help address unresolved issues of equity.

## 2. Background

### 2.1. Immigrant Populations in the United States

The concept of Shared Language Erosion is important for developmental and family scientists given the growth of new immigrant areas in the US. According to the 2019 Current Population Survey, immigrants and their children represent 28 percent of the US population [18]. The US Latino population, in particular, has increased by about 2 percent a year for the past decade, growing from 50.7 million in 2010 to 60.6 million in 2020 [19] with no anticipated abatement. It is projected that 88 percent of the US population growth over the next 50 years will be due to immigrants and their offspring [20]. Thus, there is an urgent need to better identify and understand key factors that affect and shape immigrant families, and to translate such understanding into culturally appropriate and relevant programs, interventions, and public policies.

Today, the increased visibility of the immigrant population in the US is largely due to the population growth of children of immigrants. As of 2018, an estimated 18 million children under 18 years of age lived with an immigrant parent in the US [21]. Data from the US Current Population Survey regarding languages spoken at home reveals a substantial increase in dual lingual households over the past three decades. In 1990 14 percent of children aged 17 and under spoke a language other than English at home; this percentage rose to 23 percent in 2019 [18,22]. The increase in the number of children in immigrant families who speak a language other than English in the home has pressured education systems to accommodate ELL (English Language Learner) students and has ushered in the formation of ELL and dual language programs. ELL students comprised a surprising 10.1 percent of all students in the US public school system in 2016 [23].

The communities into which immigrant families settle may influence their educational trajectories and other outcomes such as physical, emotional, and behavioral health. For example, the shift in Latino migration from long-standing enclaves like Florida, Texas, Arizona, and California to the Midwest has grown dramatically since 1990, and has created “new settlement” areas where there are few established immigrant populations [24,25]. Compared to long-standing immigrant enclaves, the human and social services infrastructures of new settlement areas are relatively poorly equipped to meet the needs of rapidly growing immigrant populations [26]. Frequently, in these areas, programs that provide services (e.g., social, health, education) have not been translated or culturally adapted to meet the unique needs of immigrants [27,28].

Even with its vast numbers of immigrants, the US has been described as a “graveyard” for languages [17,29]. Demographic research shows a stark decline in the use of HLs among immigrant families across the US [17,30,31]. Even among Latino immigrants (who are the strongest maintainers of their HL), there are substantial documented decreases in Spanish use from the first to the second and third generations [17]. Overall, the majority of third-generation immigrant children speak only or mostly English in the home, which implies that they are not likely to develop bilingual language skills [30,31]. The decrease in HL use across generations indicates that Shared Language Erosion is likely a widespread phenomenon in the US. Additionally, Shared Language Erosion is likely to be more pronounced in smaller and less established immigrant communities as there are fewer opportunities for children to develop and maintain their HL and fewer formal opportunities for parents to learn English.

### 2.2. Language as an Essential Component of Acculturation

Language use is only one component of the various challenges and possibilities that immigrants face; they live in a space between cultures and must grapple with the adoption of values, beliefs, language(s), and behaviors of the host culture while maintaining certain aspects of their heritage culture through a process known as acculturation [32,33]. A sizeable literature has demonstrated that children tend to acculturate more quickly than adults, which can lead to acculturation “gaps” or discrepancies in cultural orientations [33,34,35]. However, findings on the effects of acculturation are mixed (see [13] for a review). For instance, some work finds no significant link between parent-child acculturation differences and family conflict and child outcomes [36], while others found that high parent-child relationship quality moderated the association between acculturation differences and parent-child conflict [37].

Overall, whereas different rates of adapting to a new culture can create “gaps” in parent and youth beliefs, values, and behaviors, there may be another factor influencing whether these “gaps” result in conflict that affects child outcomes. To this point, a handful of studies using self-reported language proficiency have suggested that Shared Language Erosion increases parent-child conflict [38,39]. We propose that the traditional focus on parent-child differences in cultural values, beliefs, and behaviors may obfuscate the more fundamental explanation of parent-child conflict stemming from miscommunication due to a lack of shared common language.

### 2.3. Family Language Policy

In fully bilingual families, it is common for family members to effortlessly switch from one language to another [40,41]. However, achieving family bilingualism requires effort on the part of all family members. Recent research on bilingualism has focused on what is referred to as family language policy, or parental attitudes and goals concerning heritage and/or host language use in the family (e.g., [42,43,44,45,46]). The majority of studies on home and parental language use indicate that the extent to which immigrant families use English in the home has positive impacts for children’s English skills, but negative impacts for their HL skills [47,48,49]. In contrast, initial studies suggest that HL use at home has no effect on children’s English acquisition but may have a small but statistically significant effect on children’s HL acquisition [50,51]. Therefore, the specific use and practice of a HL supports children’s initial HL development, whether it is between parents and children [51,52,53], parental use of the HL with each other [45], or communication with extended kin and the HL community (e.g., [50]). However, when siblings act as sources of the host language for each other, the opposite occurs [45,54]. Siblings’ exposure to the school environment where English is the main language spoken quickly leads to English as the preferred language in sibling interactions. As siblings increasingly interact in English, the family’s language policy changes to decreased use of the HL, and results in Shared Language Erosion, particularly for younger siblings.

Family language policy also has important implications for family relations. Families who share a HL report more harmonious relations and cohesion [46,55,56,57,58], respectful adolescent-parent relationships [59] and less family conflict [9]. In fact, fluent bilingualism among family members has been shown to result in the most beneficial family relations relative to other configurations of family language use [12,60,61]. It appears that bilingualism, as a family language policy, allows children to adapt to and interact freely with the host culture, which promotes child well-being [61], positive psycho-social adjustment [12,62], self-esteem [63], and secure attachment patterns [64], while preserving a strong ethnic identity [65,66,67]. In contrast, when adolescents are not proficient in their HL, they may feel detached from it, and therefore, less connected to their parents, their extended kin, and their HL community.

### 2.4. Education Impacts on Language Learning and Use

Notwithstanding the importance of family language policy on adolescent children’s HL development and maintenance, the data suggest that achieving comprehensive HL proficiency is unlikely without formal language instruction [48]. Unfortunately, school-based bilingual language educational programs are often either not available or economically inaccessible to immigrant families. This is poignantly true in new settlement areas where culturally and linguistically appropriate resources may be scant. This is a significant deficit considering that bilingual language programs have proven to be incredibly beneficial, not only for young children’s maintenance of their HL, but also for the development of their English language skills [48,68,69]. Although dual language programs are increasing in the US (from 260 in 2000 to more than 2000 in 2011 [70]), they remain unevenly distributed across the country [71]. Even in areas that do have dual language programs, age-appropriate proficiency levels in both languages can lag because classrooms frequently do not use both languages equally [48]. The negative ramifications for immigrant children not receiving a bilingual education in early childhood are known; children who are instructed exclusively or predominantly in English show deteriorating HL scores [72], which increases their risk for experiencing Shared Language Erosion and family dysfunction.

### 2.5. Language Use in Practice

One major benefit of dual language instruction is that it increases the number and diversity of domains in which an immigrant child becomes proficient in their HL. The purpose of school is to expose and instruct children in different domains of knowledge that are essential for their capacity to function in society. For immigrant children, however, the exposure to these domains occurs mainly in English, without corresponding exposure in their HL. This results in rapid and dramatic growth in the child’s ability to comprehend and express themselves in English and an equally noticeable reduction in the array of areas around which they can freely and effectively interact with others in their HL [16]. For example, even though some second- and third-generation immigrant adolescents are able to *speak* their HL, their HL literacy skills (e.g., reading and writing) are not as proficient, suggesting a non-comprehensive command of their HL [17].

Although there is scant research on domain-related language use with objective measures, it is logical to presume that limited exposure to diverse domains of use will result in fewer opportunities for adolescents to practice their HL, and, therefore, they will have mastery of a narrower range of linguistic forms and concepts in that language [16]. This has implications for parent-adolescent relationships: studies find the quality of parent-adolescent relationships is not related to *whether* adolescents speak the HL with their parents, but rather, to the degree of proficiency that adolescents have in their HL [73]. Therefore, it is not just being able to speak a HL that fosters positive outcomes for adolescents, but rather the *matching* of parent-adolescent HL proficiency [60]. Further contributing to Shared Language Erosion is adolescent host language development. Because immigrant children learn English in school (e.g., [74,75], the domains in which they have English language competency are likely to be quite different from their parents, who may learn English through their employment or in other locations.

Matching HL proficiency also varies across the assorted domains in which parents socialize their children. Socialization is the “process by which children acquire the social, emotional, and cognitive skills needed to function in the social community” [76] (p. 691). Parental socialization is complex enough within one’s home culture and is often exponentially more difficult when parents and children have the added burden of adapting to a host culture. It is increasingly recognized that how parents socialize their children differs across distinct social domains [76,77]. For instance, adolescents may more freely express their autonomy in one domain (e.g., personal choices such as music and clothing preferences), while desiring more autonomy in other domains like the prudential domain (e.g., involvement in activities parents consider risky [77]). Thus, the domain that is most challenging for the parent-adolescent relationship may be the area in which shared language erosion is the greatest obstacle for productive communication. To constructively negotiate and redefine their relationships, parents and adolescents must be able to communicate effectively within these different domains. To the extent Shared Language Erosion hinders their ability to communicate, the relationship may suffer.

Differences in language fluency in immigrant families can lead to issues with communication, especially arriving at intersubjectivity. For example, qualitative work on negotiating understanding in immigrant family interactions shows that linguistic and cultural misunderstandings arise throughout everyday family interactions [78]. Interlocutors adjust and modify their language use to orient to each other’s lack of expertise via checking their understanding before continuing a conversation, modifying their own talk for the benefit of a novice speaker, or assuming competency and then having to revise their talk when it becomes apparent that someone else has misunderstood. This study shows that family members appear to be aware on some level of their other family members’ linguistic and cultural competencies and deficiencies and may account for these deficiencies in the design of their talk. However, this study also notes how people sometimes ‘get it wrong’ by assuming competency or lack thereof. We propose that these intercultural misunderstandings created by Shared Language Erosion create risk for conflict in parent-adolescent relationships.

Shared Language Erosion is likely to occur when parents and adolescents do not match across various domains of language use. Although home language use promotes HL maintenance to some extent, it is insufficient to create age-appropriate fluency. When children are not given ample opportunities outside of the home to practice their HL, there are negative implications for their development and/or maintenance of the HL. In other words, attaining comprehensive proficiency in a HL is unlikely to occur unless there is formal language instruction where adolescents are able to achieve competency in all components of a language (e.g., appropriate semantics, syntax, pragmatics) in a variety of domains (e.g., about home-related affairs but also current events, emotional and relational contexts, etc.). To the extent that adolescents are not exposed to their HL in multiple domains, a preference for English quickly develops and begins to replace use of the HL even in the home with their relatively monolingual parents.

## 3. Effects of Shared Language Erosion on Immigrant Families

In what follows we propose that Shared Language Erosion can impact family functioning and adolescent development in three primary ways, through: (a) increases in parent-child conflict, (b) reductions in parental competence that influence the family’s ability to respond to external and internal stressors, and (c) the aggravation of any preexisting deficiencies in parent-child attachment. Any one of these or the combination of two or more of them can leave the developing adolescent vulnerable to environmental risks such as delinquent peer groups, thereby starting or continuing a negative developmental cascade that impacts their physical, emotional, and behavioral health.

### 3.1. Shared Language Erosion and Parent-Child Conflict

First, perhaps the most noticeable effect of Shared Language Erosion is parent-adolescent conflict. In our current research, over 40 percent of Latino immigrant adolescents report high levels of misunderstanding in their communication with their parents due to navigating two languages at home, and their parents report similar numbers [79]. Adolescence is a developmental period known for parent-child conflict (e.g., [80]). Notably, though, conflict between adolescents and their parents most often involves mundane topics such as daily hassles resulting in either negative or neutral affect [81]. These mundane everyday disagreements do have some positive implications: they allow for adolescents and their parents to negotiate expectations, roles, and responsibilities to accommodate the increasing autonomy of the developing adolescent [82]. However, with Shared Language Erosion, engaging in these types of disagreements constructively may prove difficult since open communication between parents and adolescents is conditioned particularly on parental understanding [83], which, in turn, is critical to intersubjective understandings of expectations, roles, and responsibilities. Some researchers point to this issue with young children, writing that families who do not learn to communicate effectively run the risk of dysfunctional discord during adolescence due to the developmental changes of adolescents seeking autonomy during this stage of life [84].

As the control domain (parental authority) gets challenged during adolescence, there is often negotiation and compromise as parents and adolescents try to find a mutually acceptable middle ground. This process varies across ethnicities and across generations of immigrants [85]. For example, first-generation immigrant adolescents are more likely to fully disclose their personal lives to their parents than subsequent generations [85], this behavior is associated with fewer behavior problems in new immigrant youth. However, Shared Language Erosion is likely to hinder parents’ and adolescents’ ability to understand each other and thereby reach optimal resolutions to conflict, even if there is adolescent disclosure.

### 3.2. Shared Language Erosion and Parental Competence

Second, we propose that the erosion of a shared language affects parental ability to communicate their life experiences and wisdom to their adolescents and/or their ability to adequately monitor (e.g., [86,87]) their adolescent children’s behavior. In other words, parenting may be affected by lack of proficiency in a shared common language. Prior research on generational dissonance (when there are parent-child differences in terms of cultural orientation), which is linked to decreased supportive parenting and adolescent depressive symptoms, points to the potential for language to be a core component of this dissonance [35,88]. For example, when parents were minimally proficient in English and children were minimally proficient in the HL, these dyads had the least supportive parenting and highest adolescent depressive symptoms [88]. However, adolescents who are more fluent in their heritage language report higher levels of parental warmth [89]. These findings indicate that not having a shared language is a major contributor to issues with immigrant parenting success. Other work on language has similar findings; adolescents who spoke in a different language than their parents reported less cohesion and discussion with them than adolescents who spoke the same language as their parents [58]. This may be due in part to parental desires; parents often want their children to be fluent in their HL [90].

Shared Language Erosion directly affects how well parents understand their child’s perspective, which is crucial for mutual understanding and cooperation. Understanding adolescents’ thoughts and feelings during a conflict predicts better outcomes [91]. Knowing how children will respond to different disciplinary tactics is associated with more cooperation from them [91]. Parents who feel less competent in controlling their children are more likely to feel threatened and therefore resort to overly punitive authoritarian tactics [92]. Research on effective parenting suggests that not having a shared language can have negative impacts on parenting styles. Authoritative parenting has been shown to be optimal for children’s academic, cognitive, social, and behavioral outcomes [93] and results in less conflict between parents and children [94]. Authoritative parenting also affords more child self-disclosure [94], which is beneficial for parent child-relationships [95]. Shared Language Erosion can exacerbate a decrease in authoritative parenting, pushing parents towards overly lax or punitive parenting, and discourage adolescents from engaging in self-disclosure.

Effective parenting behaviors such as parental monitoring may also be affected by Shared Language Erosion. For instance, researchers found that effective parenting behaviors (e.g., monitoring) that are generally associated with positive outcomes in adolescents did not produce those results in families where there was not a common language between mothers and adolescents [39]. That is, having a shared common language is the difference between traditionally effective parenting behaviors having a positive impact on adolescent outcomes such as substance use. The effect of Shared Language Erosion can also be seen in parental school involvement. When adolescents are more fluent in English than their mothers, maternal school involvement is significantly lower than when both are proficient in either English or Spanish [96]. That is, parental proficiency in the host language is not as important in determining parental involvement in their children’s education compared to a matching of parent-child language proficiency. Not speaking the same language(s) with the same level(s) of proficiency as one’s adolescent children hinders immigrant parents’ abilities to parent effectively, resulting in a heightened probability for their adolescents to participate in risky behaviors.

### 3.3. Shared Language Erosion and Attachment

Third, Shared Language Erosion may also impact the trajectory of the attachment process. In other words, the erosion of a shared language in adolescence may exacerbate the effects of less-than-optimal parent-child attachment [97] in the early developmental period. The importance of parent-child attachment during adolescence is controversial in comparison to its importance during earlier developmental stages. Some studies find that attachment models are relatively fixed throughout adolescence; having securely attached relationships during early childhood positively impacts later relationships (e.g., [98,99,100]. Other studies indicate that experiences during adolescence interact with early attachment relationships, and that changes in attachment security in adolescence are related to the presence or absence of negative life events [101,102]. As previously discussed, immigrant families often face unique challenges related to the process of acculturation. Shared Language Erosion may hinder the maintenance of attachment security in immigrant adolescents.

Secure attachment during adolescence has positive effects for adolescents’ communication with people in their lives. Adolescents who are securely attached to one or both parents report more positive and fewer negative interactions with their parents [103] and better-quality daily interactions with people other than their parents [104]. There are also clear connections between language and attachment: adolescents who prefer their HL have indicators of secure attachment patterns [3,64]. We can assume that the reverse may also occur when adolescents have indicators of insecure attachment patterns: Shared Language Erosion may exacerbate small flaws in what normally would be “good-enough” attachment.

One key characteristic of adolescence is that adolescents begin to distance themselves from their parents as they spend more time with peers and seek more autonomy. Therefore, other relationships (i.e., peer relationships) impact one’s attachment security during adolescence [105]. In particular, adolescents are more likely to turn towards their peers to fulfill attachment functions if they view their relationship with their parents as less secure [106,107]. As the process of Shared Language Erosion aggravates pre-existing fissures in adolescent-parent attachment, it likely pushes adolescents to further rely on their peers for attachment security.

### 3.4. Shared Language Erosion and Peer Influence/Selection

Finally, any one of the effects of Shared Language Erosion on parent-child relationships may leave the adolescent more vulnerable to environmental risks such as delinquent peer groups. The connection between peers and negative/risky behaviors in adolescence is well documented (e.g., [108]). Although many models have been used to explain peer relationships and negative behaviors, one of the most fully developed is Coercion Theory, which posits that consistent negative interactions in early and late childhood reinforce aggressive behaviors and the selection of deviant peers as children move into adolescence [109]. The results of parental coercion and subsequent conduct problems in classroom settings (e.g., [110]) can result in children being rejected by more prosocial peers, prompting a cascading effect towards further antisocial behavior with deviant peers (e.g., [108,111]). Parents who have limited means for communicating effectively with their adolescent children due to Shared Language Erosion may be more likely to use coercive parenting strategies: aversive behaviors used contingently as a means of controlling the behavior of another [109]. Thus, Shared Language Erosion may add to or intensify negative interactions within immigrant families and therefore lead to increased risk for negative outcomes in adolescence through peers [112].

Since adolescents have a strong desire for peer belonging, the language attitudes and language behaviors of their peers also influence immigrant adolescents’ proficiency in both English and their HL [41]. Having peers who speak the same HL also appears to be beneficial: association with peers who have strong HL skills has been found to buffer against the loss of HL skills [113], even if it does not result in a strong ethnic identity [114]. We extrapolate that if adolescents do not speak the same language with the same level of proficiency as their parents (i.e., they are losing or not developing their HL), they are more susceptible to negative peer influences. For instance, immigrant childhood language has been found to be an early marker of substance use trajectories: higher levels of English use in childhood are associated with increasing alcohol and tobacco use trajectories, and lower levels of Spanish use are associated with increasing cannabis use trajectories [115]. Similarly, research finds that discrepancies in parent-child English language ability are associated with increases in adolescent alcohol use [34].

## 4. Discussion

In this paper we laid the foundation for a new concept that we are calling Shared Language Erosion (SLE) in immigrant families. Shared Language Erosion is based on the observation that among immigrants to the US, second-generation children not only acquire English at an accelerated rate, but also lose proficiency in their HL. Concurrently, the first-generation parents of such children make modest gains in the use of the English language while maintaining their HL. Thus, the language that parents and their children previously shared erodes.

As previously described, erosion of a shared language impacts immigrant families in three primary ways. First, because adolescents generally maintain enough of the HL to maintain a basic level of communication with their parents (and their parents gain some levels of English language competence), neither the parents nor adolescents are necessarily aware of the potential for decreased intersubjectivity during everyday communication. Therefore, Shared Language Erosion creates conflict in the parent-adolescent relationship due to linguistic and cultural misunderstandings, and does not allow for effective reconciliation which is vital in adolescent development of autonomy. Second, the erosion of a shared language negatively impacts parents’ ability to communicate their life experiences and wisdom to their adolescent children, resulting in decreased parenting effectiveness and ability to monitor their children’s daily activities. From this perspective, the erosion of a shared language may increase parent-child conflict and reduce parental competence. Third, because Shared Language Erosion dampens parent-child communication, it may serve to exacerbate pre-existing issues (e.g., poor attachment) in the parent-child relationship. Finally, any one of the effects of Shared Language Erosion on parent-child relationships may increase the vulnerability of adolescents to environmental risks such as delinquent peer groups.

One major potential contribution of Shared Language Erosion to the existing literature on immigrant experiences is a potential explanation for the immigrant paradox, in which first-generation immigrants often outperform more established immigrants and non-immigrants on various health-, education-, and crime-related outcomes [8]. Several explanations have been put forth to account for this paradox, the most popular of which is acculturation. We posit that Shared Language Erosion can help clarify the mixed findings of the impact of acculturation on parent-child relationships. As previously noted, research on acculturation often includes several aspects of life (e.g., language, values, beliefs, norms, food and music preferences) to measure participation in one culture or another. However, we want to make clear that our view does not see language simply as one indicator of participation in or belonging to a culture that contributes to the immigrant paradox. Our view is that because culture is socially constructed (i.e., we socially create meanings and culture through communication) [116], communication, and the language(s) used to communicate, are at the core of parent-child relationships and shared cultural identity. Furthermore, the erosion of a shared language between parents and their adolescent children slowly and subtly robs them of their ability to achieve intersubjectivity, which in turn leads to the documented increases in deviant behaviors between first- and second-generation immigrant youth. In other words, Shared Language Erosion is likely to be the greatest in second (and possibly third) generation immigrants, especially compared to the first generation, and helps to explain why immigrants are more at risk for problematic health outcomes in the second and third generation.

Furthermore, research in neuroscience shows increasing evidence for the role of environmental factors such as stress in the transgenerational transmission of neuronal and behavioral adaptations mediated by epigenetic mechanisms [117]. It may be that the chronic stress and conflict resulting from ineffective parent-child communication predisposes children to engage in behaviors that replicate the relationships they experienced with their parents with their own children. Regardless of the presence of epigenetic mechanisms that predispose parental behaviors, a substantial body of evidence has documented the transmission of parenting behaviors from one generation to subsequent generations (e.g., [118]). Although to date no such evidence links the experience of Shared Language Erosion in adolescence to subsequent parenting behaviors, it is not a leap to hypothesize how this might happen. In fact, various theoretical perspectives propose as much (e.g., [97,119,120], even if they differ somewhat regarding the presumed mechanisms through which the transmission occurs. This suggests that Shared Language Erosion may help account for the decline in physical, emotional, and behavioral health outcomes from first- to third-generation immigrants.

### Future Directions

In order to understand the scope and impact of Shared Language Erosion, there is a need for longitudinal studies on adolescent (and parent) language proficiency, especially in non-enclave immigrant areas. Although a growing body of research documents the importance of bilingual language programs on very young children and their development of host and HLs, research on adolescent language maintenance and attrition is limited. Furthermore, studies on immigrant language use are often relegated to examining children *or* parents, and to our knowledge there is no longitudinal work on language proficiency that includes both parents and children. Examining adolescent *and* parent language proficiency over time will enable us to see the relationship between their language maintenance and/or attrition and outcomes such as conflict and participation in deviant peer groups. Additionally, one possibility to explore is the shape of Shared Language Erosion over time. Some research points to the notion that it may be curvilinear because the association between language use and parent-child relationships is established in early adolescence and then becomes somewhat more stable during later adolescence [58].

Importantly, when examining Shared Language Erosion, we propose a two-fold approach to assessing language proficiency. First, there is a need for objective measures when assessing language proficiency (rather than relying on self-reports, for example). A concern regarding self-report is that it does not assess key issues related to language proficiency of which speakers themselves may not be cognizant. For example, adolescents may consider themselves to be fluent in their HL. However, they may only be able to speak with vocabulary and grammatical constructions that are relevant to and frequently utilized in their home settings. In other words, adolescents may know that they can carry out a conversation with their family members in their HL concerning home-related domains (e.g., housework, daily routines, etc.), but they may be less aware of their relative inability to converse in other domains (e.g., applying for a bank loan, talking about an emerging romantic relationship, etc.). To this point, one study found that although the vast majority of second-generation Latino youth reported being proficient in their HL, only a small percent were actually proficient when tested using a more objective measure [29]. Additionally, although parents may acknowledge that they have limited English proficiency because they only speak it in limited practical contexts (e.g., at the store, at work), their domain-specific development of English restricts their ability to hold conversations with their adolescents about home topics (e.g., relationships, the future).

Second, we propose that observing language in everyday use is an additional way to access immigrant family language fluency and discern how this affects family relationships. Although objective measures of language proficiency can tell us how proficient an individual is in their host and/or HL, this is only part of the picture of language fluency. Examining language in use is another way to ascertain how family members actually utilize different languages on a day-to-day basis. Administering self-reports of language use in the home is one way to access this, but another, more effective way may be to collect audio- and/or video-recordings of families communicating in the home (e.g., [40,78]). By observing how families negotiate language and the extent to which they are aware of potential miscommunication in real-time, it is possible to see how language fluency plays out in everyday family interactions. That is, observing how family members orient to linguistic (and cultural) knowledge is one way to see how language fluency impacts family relationships and functioning. Furthermore, examining code-switching and/or code-mixing in family interactions will be beneficial when analyzing language in use [121,122,123].

In terms of thinking about how to prevent Shared Language Erosion in immigrant families, it is important to consider potential barriers and resiliency factors related to bilingualism in new settlement areas. Although studies show that bilingual education programs are incredibly beneficial for the bilingual development of young children (e.g., [48,68,69], there is, to our knowledge, no studies that have documented these effects for children in middle and high school. Perhaps having bilingual education programs continue for older students is one key strategy to develop and maintain bilingualism for adolescents as well. One practical implication of this research would be to further develop and accredit dual language programs in middle and high schools or, at a minimum, to offer HL classes in school.

Furthermore, we also know that promoting HL use in the home has benefits for younger children (e.g., [48]). However, what about HL use in the home for older children in non-enclave locales? Similarly, it is worth noting that promoting educational attainment among immigrant parents themselves may be incredibly beneficial to combat Shared Language Erosion. In new settlement areas, immigrant parents may be more likely to speak English outside of the home (e.g., for work and shopping-related tasks), but have less access to resources for further development of their English (e.g., language schools). In other words, it may be that developing bilingual families as a whole is a key to combating Shared Language Erosion.

## 5. Conclusions

In conclusion, we do not yet know the extent to which Shared Language Erosion explains negative outcomes among immigrant youth. We propose that because language is a primary vehicle for expressing and developing one’s identity, lack of a shared language leads to adolescents having dissimilar linguistic and cultural identities as their parents, resulting in negative impacts on adolescent development and parent-child relationships. Shared Language Erosion may help explain the immigrant paradox and point to language differences as the core component of the negative impacts of monodimensional acculturation among immigrant families.

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
