# Peer review of "Shared Language Erosion: Rethinking Immigrant Family Communication and Impacts on Youth Development"

_children, 2021, doi:10.3390/children8040256_

Round 1
Reviewer 1 Report
A very convincing proposal for consolidating research in adolescent development, parenting, sociolinguistics of bi- and multilingualism.
Line 346: "sufficiently lower" - unclear
Lines 371-372: "the reverse may also occur" - spell out what "the reverse" refers to
Lines 497 - 500: The relationship between the three propositions stated in this fragment is obscured by the use of "although".
Lines 501 - 513: Unfortunately, conversation analysis used in the study by Bolden (2014) is a limited tool as it can only address cases of miscommunication and comprehension problems of which the participants are aware and which they chose to discuss. The idea of studying language is use stands and falls with the choice of adequate methodology of data collection and analysis. For example, an analysis of code-mixing and code-switching in family conversations might prove crucial to further insights. This is not a criticism pertaining to the current paper but something to consider should an empirical study of language in use materialise as proposed.
Author Response
Thank you again for your consideration of this article for the special issue of Children. The suggestions made by the reviewers were very helpful, and we believe we have addressed the minor concerns raised. We outline the changes made below:
- Lines 141, 388: rephrased “acculturative stressors” to “acculturation” which we subsequently define, as suggested by Reviewer 1
- lines 223-225: restructured this paragraph so that all Heritage Language (HL) discussion occurs first, then host language (also confirmed consistency throughout the document, as suggested by Reviewer 1
- lines 239-240, 480-483: added in references to the constitutive approach of communication to address the dynamic and constructed nature of culture (lines 469-471), slightly rephased some sentences that contain phrases such as "learning a new culture", as suggested by Reviewer 1
- line 321: corrected typo to hierarchical
- line 369: changed sufficiently to significantly to clarify meaning, as suggested by Reviewer 2
- lines 412-422: defined coercive parenting strategies, as suggested by Reviewer 1
- lines 469-471: added in references to the constitutive approach of communication to address the dynamic and constructed nature of culture as suggested by Reviewer 1, slightly rephased some of the sentences that contain phrases such as "learning a new culture"
- lines 532-535: revised sentence for clarity, as suggested by Reviewer 2
- lines 564-655: added in some additional citations and further methodological considerations for future research specific to code-switching, as suggested by Reviewer 2
Thank you very much for taking the time to review this paper again, and we look forward to hearing your response.
Reviewer 2 Report
This is an exceptionally well-written and engaging paper. The authors discuss a novel concept, Shared Language Erosion. They provide a clear introduction and useful background information on immigrant families in the US. Their discussion of the effects of SLE on Immigrant Families is comprehensive and well articulated, and they put forward three primary ways in which SLE can impact family functioning and adolescent development. This makes an important contribution to several research fields including acculturation studies and family language policy research. The future directions for research outlined at the end are important ways forward and provide a useful ‘roadmap’ for researchers in the fields of applied linguistics, cross- and intercultural communication studies, and those researching internationally mobile individuals more broadly. I feel that this paper will be of great interest to the readership of the journal. No major changes are required, but the authors might consider approaching the notion of 'culture' from a more critical perspective. Some of the terminology used, e.g. ‘learning a new culture’ (237), 'participation in one culture or another' (444), and the terms 'home' and 'host' culture imply a fairly simplistic, static, view of culture, although I'm sure the authors are aware of the complexities surrounding this concept. I wonder whether a constructivist perspective that views cultural identities as emergent, dialogically constructed and multiple should be acknowledged here. Some other minor issues:
-
Under 2.2 the authors use the heading 'Language as an essential component of acculturative stress', but acculturative stress is not sufficiently unpacked here. Perhaps a definition could be provided.
- 364: the authors mention ‘unique acculturative stressors’ but I don’t think this is sufficiently discussed elsewhere. What exactly are these? (see point above)
-
Lines 221-232: the authors should check whether they mean HL as in 'heritage language' or 'host language'. In line 225 ‘host language development’ is mentioned but then they go on to discuss heritage language proficiency from line 228. Double-check acronyms and what they refer to.
-
394: The authors refer to coercive parenting strategies. This could be unpacked for readers not familiar with this term, and perhaps examples given.
-
Line 300: hierarchical rather than ‘hierchical’?
Author Response

(The authors gave the same response as above.)
